# Less Is More: Selective-Atom-Removal-Derived Defective MnO_x_ Catalyst for Efficient Propane Oxidation

**DOI:** 10.3390/nano14110907

**Published:** 2024-05-22

**Authors:** Wenfan Xu, Limei Zhou, Lining Liu, Huimei Duan, Haoxi Ben, Sheng Chen, Xingyun Li

**Affiliations:** 1State Key Laboratory of BioFibers and Eco-Textiles, Institute of Materials for Energy and Environment, College of Materials Science and Engineering, Qingdao University, Qingdao 266071, China; yanwu003x@foxmail.com (W.X.); lln2584038492@163.com (L.L.); 2Chemical Synthesis and Pollution Control Key Laboratory of Sichuan Province, China West Normal University, Nanchong 637002, China; cwnuzhoulimei@163.com; 3Institute of Marine Biobased Materials, School of Environmental Science and Engineering, Qingdao University, Qingdao 266071, China; 4State Key Laboratory of Coal Combustion, School of Energy and Power Engineering, Huazhong University of Science and Technology, Wuhan 430074, China; sheng_chen@hust.edu.cn

**Keywords:** MnO_x_, defects, doping, propane oxidation

## Abstract

Defect manipulation in metal oxide is of great importance in boosting catalytic performance for propane oxidation. Herein, a selective atom removal strategy was developed to construct a defective manganese oxide catalyst, which involved the partial etching of a Mg dopant in MnO_x_. The resulting MgMnO_x_-H catalysts exhibited superior low-temperature catalytic activity (T_50_ = 185 °C, T_90_ = 226 °C) with a propane conversion rate of 0.29 μmol·g*_cat._*^−1^·h^−1^ for the propane oxidation reaction, which is 4.8 times that of pristine MnO_x_. Meanwhile, a robust hydrothermal stability was guaranteed at 250 °C for 30 h of reaction time. The comprehensive experimental characterizations revealed that the catalytic performance improvement was closely related to the defective structures including the abundant (metal and oxygen) vacancies, distorted crystals, valence imbalance, etc., which prominently weakened the Mn-O bond and stimulated the mobility of surface lattice oxygen, leading to the elevation in the intrinsic oxidation activity. This work exemplifies the significance of defect engineering for the promotion of the oxidation ability of metal oxide, which will be valuable for the further development of efficient non-noble metal catalysts for propane oxidation.

## 1. Introduction

Volatile organic compounds (VOCs), primarily emitted from the transportation, chemistry, and energy industries, are the well-known detrimental air pollutants, severely endangering human health [1]. Meanwhile, VOCs can be converted into secondary pollutants, thus generating near-surface ozone, photochemical smog, PM2.5, etc., posing severe threats to the ecological environment [2]. To achieve an efficient VOC abatement, catalyst-aided oxidation at low temperature has been perceived to be one of the most advanced approaches [3,4]. Among VOCs, propane is typified as an ultra-stable light alkane and is often used as a probe molecule to screen high-performance VOC elimination catalysts [5]. Although noble metal catalysts show high catalytic performance, the high cost, scarcity, and inferior stability greatly limit their widespread application [6,7,8,9]. It is imperative to explore cost-effective and robust alternative non-precious metal catalysts [10,11,12,13,14].

Among them, transition metal oxides represented by manganese oxides (MnO_x_) are widely studied for propane oxidation due to their abundant availability, multiple valences, excellent redox properties, adjustable structures, etc. [15,16,17]. Particularly, there has been great research interest in amorphous MnO_x_ due to their randomly arranged atomic structure and abundant coordination-unsaturated sites, which could promote reactant adsorption and facilitate oxygen mobility [18,19,20]. As reported in a previous work, the catalytic oxidation performance of MnO_x_ heavily relies on their oxygen vacancies (O_v_), which is responsible for the O_2_ trapping and transformation into active surface oxygen to enhance the intrinsic oxidation activity [21]. Meanwhile, engineering a high-valence Mn^4+^ of MnO_x_ was also deemed to be vital for the benefit of alkane adsorption and activation [22].

A significant number of studies have been devoted to engineering defective MnO_x_ catalysts for environmental catalysis application. For example, Liao et al. synthesized low-crystallinity MnO_x_ catalysts through a simple pyrolysis method, demonstrating that the increase in surface defects such as oxygen vacancies (O_v_) contributed to the enhancement in selective NO oxidation activity [23]. Dai et al. synthesized MnO_x_ catalysts with high-index facets, via an ultrasound-assisted precipitation method, proposing that the steric effect between high-index facet catalysts with reactants could enhance toluene oxidation [24]. By far, heteroatom (with discrepant atomic radius and electronegativity to Mn atoms) doping in transition metal oxide has emerged as a decent way to induce crystal distortion and valence imbalance, endowing a profound defective structure and electronic state modulation [25,26,27,28]. For example, Xing et al. synthesized MnO_x_ catalysts doped with varying amounts of Sb via the co-precipitation method. Sb doping was found to enhance the catalyst surface acidity, redox ability, and chemisorbed oxygen content, thereby improving catalytic activity for the removal of NO and toluene [29]. Zhao et al. constructed Ce-doped MnO_x_ using a redox precipitation method, which demonstrated significantly increased O_v_ and active oxygens for efficient toluene oxidation [30]. Wang et al. unveiled the efficacy of Fe doping in MnO_x_ for the stimulation of oxygen mobility and the increase in Mn^4+^ ratio to achieve efficient VOC oxidation [31]. Chen et al. developed a reflux strategy to prepare a Zr-doped amorphous MnO_x_ catalyst, which displayed a remarkable redox property and augmented acidity, embodying an improved performance for the oxidation of chloride containing alkanes [32]. Moreover, the promoting effect of alkaline metal doping in MnO_x_ has been successfully validated for toluene oxidation aided by the reinforcement of reducibility and CO_2_ desorption [33]. Despite the above endeavors, there is still a continuous demand for the structure manipulation of MnO_x_ to advance the practical application in low-temperature VOC oxidation [17]. Based on the recognition of the importance of defect engineering by metal doping, a rational reasoning has been suggested regarding whether further structure tailoring could be achieved by selectively removing part of the dopant. Therefore, it is expected that the metal vacancies will be occupied, leading to local crystal defects, promoting O_v_ formation, and increasing Mn valences; this strategy, however, has not been explored for propane oxidation.

Herein, an advanced defect engineering approach was proposed by selectively removing the Mg atoms in the Mg-doped MnO_x_. The “Less is more” function was realized by Mg de-doping, prominently stimulating the catalytic oxidation activity of propane and hydrothermal stability. With systematic characterizations, the structure–performance co-relationship was established. This work may pave a new path for the manipulation of high-performance non-precious metal catalysts in the application of VOC oxidation.

## 2. Experiment

### 2.1. Materials

The following compounds were used: KMnO_4_ (Sinopharm Chemical Reagent, Shanghai China), Mn(Ac)_2_·4H_2_O (Sinopharm Chemical Reagent, Shanghai China), Mg(Ac)_2_·4H_2_O (Sinopharm Chemical Reagent, Shanghai China) and HNO_3_ solution (Sinopharm Chemical Reagent, Shanghai China), respectively. All thechemicals used in this study are reagent grade with higher than 99.0% purity.

### 2.2. Catalyst Preparation

MgMnO_x_: Mg-doped MnO_x_ was prepared via a co-precipitation procedure. Typically, 1.85 g of KMnO_4_ was dissolved in 100 mL of deionized (DI) water with vigorous stirring to form a homogeneous solution. Then, 4.3 g of Mn(Ac)_2_·4H_2_O and 0.188 g of Mg(Ac)_2_·4H_2_O were added into the solution under continuous stirring at room temperature for 4 h. The obtained products were collected by filtration and washed several times with DI water and ethanol. Finally, the solids were dried in oven at 100 °C for 12 h and calcined at 300 °C for 4 h in air to obtain MgMnO_x_. As a reference, MgMnO_x_ with different Mg(Ac)_2_·4H_2_O amounts of 0.038 g, 0.188 g, and 0.376 g was prepared, corresponding to MgMnO_x_-1, MgMnO_x_-2, and MgMnO_x_-3.

MgMnO_x_-H: To partially remove the Mg dopant, 0.3 g of MgMnO_x_ was treated in 150 mL of HNO_3_ solution (5 mol·L^−1^) with stirring for 5 h at room temperature, followed by filtration and washing several times, which was dried at 100 °C for 12 h. The obtained sample is denoted as MgMnO_x_-H. As a reference, MgMnO_x_ was also prepared with different acid washing times (3 h, 5 h, 7 h) to produce MgMnO_x_-H-t (t = 3 h, 5 h, 7 h).

### 2.3. Catalysts Characterizations

Details regarding X-ray Diffraction (XRD) patterns, nitrogen adsorption–desorption isotherms, Scanning Electron Microscopy (SEM), Transmission Electron Microscopy (TEM), High-Resolution Transmission Electron Microscopy (HRTEM), H_2_ Temperature-Programmed Reduction (H_2_-TPR), Raman Spectroscopy, X-ray Photoelectron Spectroscopy (XPS), Electron Paramagnetic Resonance (EPR), Inductively Coupled Plasma (ICP), and in situ Diffuse Reflectance Infrared Fourier Transform Spectroscopy (in situ DRIFTs) are provided in the Appendix A.

### 2.4. Catalytic Performance Evaluation

As the setup in Appendix A schematically shows, propane oxidation was carried out in a fixed-bed reactor using a reactant containing 0.5 *vol.*% C_3_H_8_ and 10 *vol.*% O_2_, balanced with Ar. The weight hourly space velocity (WHSV) was set at 60,000 mL·g^−1^·h^−1^ controlled by an MFC (Mass Flow Controller). The reaction temperature was increased from room temperature to 400 °C with a heating rate of 2 °C·min^−1^. The products were in situ-tested by a gas chromatograph (Shimadzu, Kyoto, Japan, GC-2014).

C_3_H_8_ conversion was acquired from the following formulas:(1)$$C_{3}H_{8}\;\mathrm{conversion}\;\left(\%\right)=\frac{{\left[C_{3}H_{8}\right]}_{\mathrm{in}}-{\left[C_{3}H_{8}\right]}_{\mathrm{out}}}{{\left[C_{3}H_{8}\right]}_{\mathrm{in}}}\;\times \;100\;(\%)$$
where [C_3_H_8_]*_in_* and [C_3_H_8_]*_out_* represent the inlet and outlet C_3_H_8_ concentrations, respectively.

To obtain information of the apparent activation energy (E_a_), the reaction was controlled in the kinetic regime (C_3_H_8_ conversion under 10%). The equation below was used for the calculation of E_a_ by acquiring the slope:(2)$$\mathrm{lnk}=\frac{{-E}_{a}}{\mathrm{RT}}+C$$
where k is the reaction rate, T stands for the absolute temperature, and R represents the molar gas constant.

## 3. Results and Discussions

TEM and HR-TEM in Figure 1a,d show that MnO_x_ has an overall amorphous structure with localized crystals in the framework. MgMnO_x_ inherits the morphology of MnO_x_, but with a slightly looser structure (Figure 1b,e). The EDS elemental mapping (Appendix A) shows that the Mg element is uniformly dispersed in MnO_x_, indicating that Mg atoms may be successfully doped in MnO_x_. MgMnO_x_ was further etched by acid to enrich the defective structure. Compared with the MgMnO_x_ sample, the MgMnO_x_-H sample (Figure 1c,f) shows a similar morphology, indicating that acid treatment did not destroy the whole texture. The Mg contents were determined to have decreased from 0.54 *wt.*% for MgMnO_x_ to 0.17 *wt.*% for MgMnO_x_-H, as indicated by the ICP test. This demonstrates that the doped Mg atoms could be partially eliminated during the pickling process. Eventually, defect-rich MgMnO_x_-H was formed with the co-existence of metal vacancies derived from acid treatment as well as the remaining Mg dopant, which could also be indicated by the even distribution of Mg in the EDS elemental analysis in Appendix A.

XRD was performed to explore the crystal phase of all catalysts, as shown in Figure 2a. It can be found that all catalysts display peaks at 25.3°, 37.0°, and 65.8°, assigned to birnessite-type MnO_2_ (δ-MnO_2_, JCPDS No. 80-1098). The broad and weak peaks indicate the poor crystallinity of all catalysts [34]. No Mg-related species are observed for MgMnO_x_ and MgMnO_x_-H, indicating that Mg is doped into the lattice of MnO_x_ without forming bulk Mg crystals. The N_2_ absorption–desorption isotherms in Figure 2b demonstrate a typical IV-type isotherm with an H2-type hysteresis loop, indicating the existence of abundant mesopores [35,36]. The mesoporous structure is further validated by the BJH pore size analysis (Figure 2c), which indicates that the mean pore size for MgMnO_x_-H is 7.9–8.9 nm, larger than those of MnO_x_ (6.9 nm) and MgMnO_x_ (7.9 nm). The specific surface area of MgMnO_x_-H is demonstrated to be 110.3 m^2^/g, smaller than those of MnO_x_ (163.8 m^2^/g) and MgMnO_x_ (126.8 m^2^/g) (Table 1), which may be due to the larger pore size in MgMnO_x_-H. Thus, the unique large mesopores of MgMnO_x_-H will be conducive to promoting mass transfer and alleviating the competitive adsorption of water molecules on active sites, thus benefitting the hydrothermal stability improvement.

XPS analysis was employed to investigate the surface information of the catalyst. From Figure 3 and Table 1, it can be seen that Mn 2p_3/2_ XPS spectra can be deconvoluted into two peaks at 642.2 eV and 643.4 eV, corresponding to Mn^3+^ and Mn^4+^ species. As shown in Figure 3a and Table 1, the Mn^4+^/Mn^3+^ ratio of MgMnO_x_ (1.21) is higher than that of MnO_x_ (0.95). The increase in Mn valence state may be due to the partial substitution of divalent Mg for trivalent Mn ions. The Mn^4+^/Mn^3+^ ratio of MgMnO_x_-H (1.12) slightly decreases after acid etching but is still higher than that of the original MnO_x_. Furthermore, Mn 3s XPS spectra (Figure 3b) were deconvoluted to shed light on the AOS of Mn. As shown in Table 1, the AOS values for MnO_x_, MgMnO_x_, and MgMnO_x_-H are 3.40, 3.48, and 3.43, respectively. The change in AOS variation coincides with the trend of Mn^4+^/Mn^3+^ ratio variation. In addition, the O 1s spectra (Figure 3c) were fitted to two peaks at 529.8 eV and 531.5 eV, assigned to surface lattice oxygen (O_latt_) and surface adsorbed oxygen (O_ads_) species, respectively [37,38]. Interestingly, the O_ads_/O_latt_ ratio of MgMnO_x_ (0.40) increases after doping Mg in MnO_x_ (0.32), indicating that the partial substitution of Mg for Mn ions generates more surface O_v_. Furthermore, the O_ads_/O_latt_ ratio of MgMnO_x_-H (0.57) continues to increase after acid etching. This indicates that the partial *de*-doping of Mg further induces the formation of more O_v_, which renders the gas-phase O_2_ easy to activate and convert to O_ads_, facilitating the deep oxidation of VOCs [39,40].

The reducibility of the metal oxide catalysts is closely associated with the mobility of surface reactive oxygen species and can be characterized by the H_2_-TPR. As shown in Figure 4a, two main reduction peaks at 260–300 °C and 300–400 °C can be clearly observed in all catalysts, corresponding to the sequential reduction of MnO_2_→Mn_3_O_4_→MnO [41]. By comparison, the low-temperature reduction peak follows MgMnO_x_-H (233.7 °C) < MgMnO_x_ (245.6 °C) < MnO_x_ (255.8 °C). This indicates that low-temperature oxygen mobility is improved after Mg doping and is further enhanced by the subsequent Mg *de*-doping. Raman spectra in Figure 4b show the peaks at 640 cm^−1^ and 350 cm^−1^, corresponding to the symmetrical vibration peak A_1g_ (v_s_) of the Mn-O bond and the E_g_ vibrational peaks of the asymmetric stretching of Mn-O-Mn [42]. A distinct Raman peak deviation is evidenced for MgMnO_x_-H compared with MgMnO_x_ and MnO_x_. This prominently suggests that more crystal defects (lattice compression, crystal distortion, etc.) are incubated for the acid-treated sample. Moreover, the force constant (*k*) of the surface Mn-O bond could be calculated based on Hooke’s law by the following equation [32,43]:(3)$$\omega =\frac{1}{2\Pi c}\sqrt{\frac{k}{\mu }}$$
where ω is the Raman shift (cm^−1^), c is the speed of light, and μ is the effective mass of the Mn-O bond. As illustrated in the inset of Figure 4b, the force constant *k* of MgMnO_x_-H (293 N/m) is smaller than those of MgMnO_x_ (299 N/m) and MnO_x_ (301 N/m), implying that Mn-O could be greatly activated to engender a weaker Mn-O bond strength, which could facilitate O_v_ formation. The defect structures of the MnO_x_, MgMnO_x_, and MgMnO_x_-H catalysts were further measured using the EPR technique. As shown in Figure 4c, all the catalysts present a distinct feature at g = 2.003, stemming from the O_v_ filling with active oxygen species (i.e., O^2−^ and O^−^) [44,45]. Obviously, the MgMnO_x_-H catalysts possess prominently more defective O_v_ than those in MnO_x_ and MgMnO_x_ catalysts. This reflects its superior redox ability of MgMnO_x_-H catalysts, which agrees well with the above H_2_-TPR and Raman results.

For the catalytic application in propane oxidation, the influence of Mg dopant amount was first scrutinized. As shown in Appendix A, the Mg doping amount follows a volcanic curve with catalytic performance, with MgMnO_x_-2 exhibiting the highest propane oxidation activity. It is speculated that excessive Mg could form MgO on the surface of MnO_x_, thereby covering part of the active sites, resulting in activity deterioration. Meanwhile, by changing the acid etching time, it was found that the MgMnO_x_ treated by acid for 5 h has the highest activity (Appendix A). Too long an acid pickling time may result in a pore structure collapse and specific surface area reduction, leading to a decrease in the catalytic performance. The optimized MgMnO_x_-H was compared with MnO_x_ and MgMnO_x_ for propane oxidation, with the result shown in Figure 5a, and detailed activity data are illustrated in Table 2. Among them, the MgMnO_x_-H catalyst displays the best catalytic activity with a remarkably lower T_50_ (temperature for conversion of 50% propane) of 185 °C compared to those of the MnO_x_ and MgMnO_x_ catalysts (T_50_ = 242 and 208 °C, respectively). Meanwhile, the reaction rate of the MgMnO_x_-H catalyst was calculated to be 0.29 μmol·g*_cat._*^−1^·s^−1^, 2.8 and 4.8 times those of MgMnO_x_ (0.10 μmol·g*_cat._*^−1^·s^−1^) and MnO_x_ (0.06 μmol·g*_cat._*^−1^·s^−1^) at 195 °C, respectively. Moreover, by evaluating E_a_ (Figure 5b) in the kinetic range (with propane conversion lower than 10%), it can be evidenced that MgMnO_x_-H has an E_a_ of 85.1 kJ·mol^−1^, significantly lower than those of MgMnO_x_ (166.1 kJ·mol^−1^) and MnO_x_ (179.3 kJ·mol^−1^). This sufficiently proves the excellent propane activation capacity of MgMnO_x_-H in achieving a superior propane oxidation activity. To investigate the influence of acid treatment, the pristine MnO_x_ was also etched by the acid with a similar condition to that of MgMnO_x_-H. It can be seen from Appendix A that MnO_x_-H shows a slight activity improvement with T_50_ reduced from 242 °C to 224 °C, which is, however, still obviously inferior to that of MgMnO_x_-H. This result strongly proves that the activity improvement for MgMnO_x_-H is more likely derived from the process of Mg atom *de*-doping.

Hydrothermal stability is another important factor in practical application. As shown in Figure 5c and Appendix A, both MnO_x_ and MgMnO_x_ demonstrate excellent stabilities under dry conditions, experiencing only a 5% and 4% decrease, respectively, in the first ten hours. However, when water was introduced, a more profound activity deterioration was evidenced with propane conversion decreasing by 11% and 8% for MnO_x_ and MgMnO_x_, respectively. Once the water was cut off, the conversion could almost be recovered. Remarkably, it is observed that MgMnO_x_-H maintains an almost unchanged 97% propane conversion under dry conditions. The introduction of 3 *vol.*% H_2_O causes a slight decrease in propane conversion from 97% to 95% over an additional 10 h of reaction time. Once the water vapor is removed, the propane conversion swiftly rebounds to 96%, showcasing stable performance throughout another 10 h of testing. The higher hydrothermal stability for MgMnO_x_-H could be possibly attributed to the unique large mesopores, which will be conducive to promoting the mass transfer and alleviating the competitive adsorption of water molecules on active sites, thus benefitting steam resistance. These findings highlight the exceptional hydrothermal stability of MgMnOx-H, indicating its potential for applications requiring resilience to moisture.

The temperature-resolved in situ diffuse reflectance infrared Fourier transform spectroscopy (in situ DRIFTS) spectra of MnO_x_, MgMnO_x_ and MgMnO_x_-H catalysts were measured to analyze the catalytic reaction mechanism of propane oxidation. As shown in Appendix A, the peaks at 2850–3020 cm^−1^ can be identified, assigned to the stretching vibration of CH_3_ and CH_2_ of adsorbed propane [43]. The peaks at 1441 cm^−1^, 1530 cm^−1^, and 1724 cm^−1^ can be attributed to the vibrational vibrations of υ_s_(C=O), υ_s_(COO), and υ_s_(CH_3_COO). The peaks at 1131 cm^−1^, 1196 cm^−1^, and 1342 cm^−1^ belong to υ(C-O-CH), υ_as_(C-O), and υ(C-O-C) vibrations, separately [46]. In addition, υ_as_(C=C) is also observed at 1429 cm^−1^, indicating that olefin is one of the transition intermediates. All the catalysts show similar reaction pathways, indicating that the propane oxidation follows similar reaction mechanisms. This shows that the oxidation of propane on manganese dioxide can be divided into three steps. Firstly, propane adsorbs onto the catalyst surface, followed by catalytic oxidation to form propanone or propene species. Subsequently, the intermediates decompose into formic acid or acetic acid, ultimately resulting in the formation of products CO_2_ and H_2_O. Generally, the conversion of propane to propylene has a low reaction energy barrier in the early stage of the reaction [47], which is favorable for the catalytic combustion of propane. MgMnO_x_-H delivered a more rapid decrease in carbonate carboxylate and olefin with the rise in temperature, which proves its higher lattice oxygen activity, accelerating the conversion of intermediates and improving the catalytic combustion performance of propane. Based on the catalyst characterizations, we can infer that the high catalytic performance could be closely related to the weakened Mn-O bond in MgMnO_x_-H, which remarkably promoted the O_v_ formation and strengthened the active oxygen mobility. Meanwhile, the increase in ratio of high-valence Mn^4+^ could also be responsible for the enhancement in propane adsorption and the following activation. Moreover, the larger mesoporous structure in MgMnO_x_-H plays an important function in facilitating mass transfer, avoiding the H_2_O coverage on the active sites and therefore ensuring an excellent hydrothermal stability.

## 4. Conclusions

In summary, we proposed a doping and partial *de*-doping strategy to obtain defective MgMnO_x_-H catalysts, which shows a superior low-temperature catalytic activity and excellent hydrothermal stability towards propane oxidation. The high catalytic performance could be due to the beneficial structures in the MgMnO_x_-H catalyst including activated Mn-O bonds, augmented oxygen vacancies, boosted active oxygen sites, enhanced oxygen mobility, higher Mn^4+^ ratios, and larger mesoporous textures. The facile defect engineering in this work will be valuable in guiding the upgrade in the catalyst for advanced VOC oxidation reactions.

## Figures and Tables

**Figure 1 nanomaterials-14-00907-f001:**
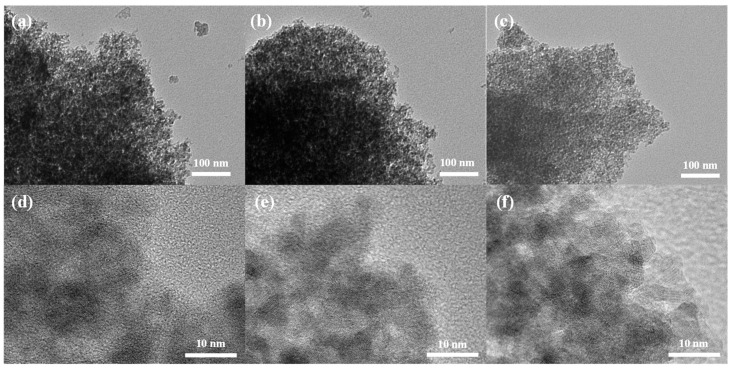
(**a**–**c**) TEM and (**d**–**f**) HRTEM images of MnO_x_, MgMnO_x_, and MgMnO_x_-H.

**Figure 2 nanomaterials-14-00907-f002:**
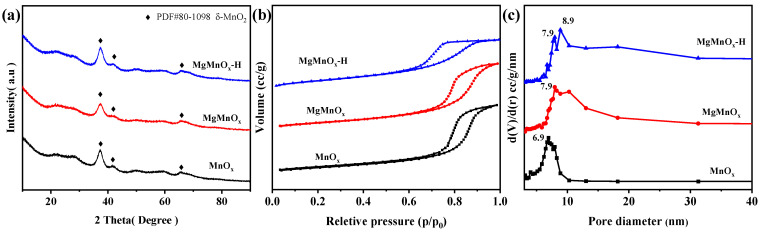
(**a**) XRD patterns, (**b**) N_2_ adsorption–desorption isotherms, and (**c**) BJH pore-size distributions of MnO_x_, MgMnO_x_, and MgMnO_x_-H.

**Figure 3 nanomaterials-14-00907-f003:**
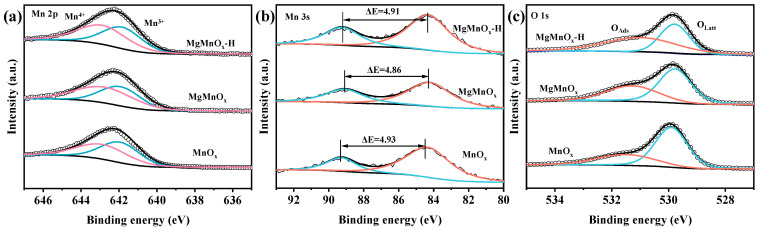
Deconvolution of (**a**) Mn 2p, (**b**) Mn 3s, and (**c**) O 1s XPS spectra of MnO_x_, MgMnO_x_, and MgMnO_x_-H.

**Figure 4 nanomaterials-14-00907-f004:**
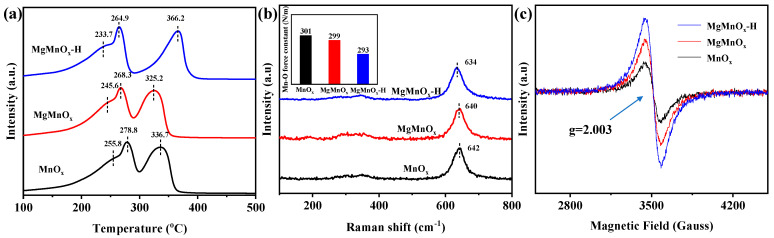
(**a**) H_2_-TPR, (**b**) Raman spectra, and (**c**) EPR spectra of MnO_x_, MgMnO_x_, and MgMnO_x_-H.

**Figure 5 nanomaterials-14-00907-f005:**
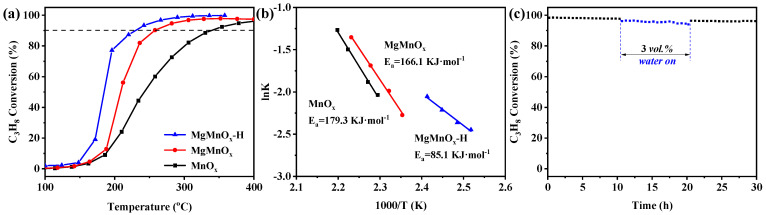
(**a**) Catalytic performance; (**b**) Arrhenius plots over MnO_x_, MgMnO_x_, and MgMnO_x_-H; and (**c**) hydrothermal stability test of MgMnO_x_-H catalysts at 250 °C (reaction conditions: 0.5 *vol.*% C_3_H_8_, 10 *vol.*% O_2_, Ar as balance gas, GHSV = 60,000 mL·g^−1^·h^−1^).

**Table 1 nanomaterials-14-00907-t001:** Pore structure and elemental information of the catalysts.

Samples	Texture Property		Mn		O	Mg
	S_BET_ ^a^ (m^2^/g)	V_t_ ^b^ (cm^3^/g)	Mn^4+^/Mn^3+ d^	AOS ^c^	O_ads_ ^d^ (%)	Content ^e^ (*wt.*%)
MnO_x_	163.8	0.27	0.95	3.40	32	-
MgMnO_x_	126.8	0.35	1.21	3.48	40	0.54
MgMnO_x_-H	110.3	0.38	1.12	3.43	57	0.17

^a^ S_BET_: specific surface area obtained by the Brunauer–Emmett–Teller method; ^b^ V_t_: total pore volume obtained at P/P_o_ =0.99; ^c^ AOS (average oxidation state) was calculated by the following equation, AOS = 8.956 − 1.126*ΔE_s_, where ΔE_s_ means the width of the interval between the two Mn 3s XPS peaks; ^d^ data calculated from Mn 2p and O 1s XPS spectra; ^e^ data obtained from ICP analysis.

**Table 2 nanomaterials-14-00907-t002:** Catalytic activities, reaction rate at 195 °C, and apparent activation energy (E_a_) of MnO_x_, MgMnO_x_, and MgMnO_x_-H catalysts.

Samples	T_50_ (°C)	T_90_ (°C)	Reaction Rate (μmol·g*_cat._* ^−1^·h^−1^)	E_a_ (kJ·mol^−1^)
MnO_x_	242	339	0.06	179.3
MgMnO_x_	208	257	0.10	166.1
MgMnO_x_-H	185	226	0.29	85.1

## Data Availability

Data are contained within the article.

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
