# Peer review of "Less Is More: Selective-Atom-Removal-Derived Defective MnOx Catalyst for Efficient Propane Oxidation"

_nanomaterials, 2024, doi:10.3390/nano14110907_

Round 1
Reviewer 1 Report
Comments and Suggestions for Authors
This manuscript presents Less is more: selective atom removal derived defective MnOx catalyst for efficient propane oxidation. In this paper, the prepared MgMnOx-H catalysts exhibited superior low-temperature catalytic activity with a propane conversion rate of 0.29 μmol·gcat.-1·h-1 for propane oxidation reaction, which is 4.8 times of the pristine MnOx. References should be updated with more recent reports about the applications of MgMnO-H catalysts. Please explain what is novel and what are the differences in this work in the introduction. The authors should clearly justify the choice of their catalysts and the advantages of their work. Please explain the difference between defective catalytic effect and metal state atomic catalytic effect.
The set-up diagram of the propane oxidation catalysis experiment should be showed for the information of distribution of the catalysts. The production rates or propane conversions using different catalysts should be presented after long time experiments. The optimization of doping masses and processes should be presented in detail.
After major revisions and clear explanations, this manuscript may be published in the Nanomaterials.
Comments on the Quality of English LanguageEnglish in some parts needs to be polished.
Reviewer 2 Report
Comments and Suggestions for Authors
This article on the development of manganese-based VOC catalysts is very well written, the methods are well consistent with each other and the results of improving the system are achieved.
There are a couple of points that are recommended to be improved -
1. In Figure 1 SEM (j) Mg for MgMnOx, it is recommended to add a granular view and mapping for other elements for this sample, since this photograph is out of line.
2. How does the Mn3+/Mn4+ ratio affect the activity of the catalyst? This ratio looks quite chaotic in Table 1.
Overall, the article looks complete and deserves publication, with minor comments.
Round 2
Reviewer 1 Report
Comments and Suggestions for Authors
The authors adjusted well according to most of my comments. The references had been updated. This manuscript can be published in the Nanomaterials now.
Comments on the Quality of English LanguageEnglish is ok.